# Self-protecting CoFeAl-layered double hydroxides enable stable and efficient brine oxidation at 2 A cm$^{-2}$

Wei Liu[1], Jiage Yu[1], Tianshui Li[1], Shihang Li[1], Boyu Ding[1], Xinlong Guo[1], Aiqing Cao[1], Qihao Sha[1], Daojin Zhou[1] ✉, Yun Kuang [2] ✉ & Xiaoming Sun [1] ✉

Low-energy consumption seawater electrolysis at high current density is an effective way for hydrogen production, however the continuous feeding of seawater may result in the accumulation of Cl$^-$, leading to severe anode poisoning and corrosion, thereby compromising the activity and stability. Herein, CoFeAl layered double hydroxide anodes with excellent oxygen evolution reaction activity are synthesized and delivered stable catalytic performance for 350 hours at 2 A cm$^{-2}$ in the presence of 6-fold concentrated seawater. Comprehensive analysis reveals that the Al$^{3+}$ ions in electrode are etched off by OH$^-$ during oxygen evolution reaction process, resulting in M$^{3+}$ vacancies that boost oxygen evolution reaction activity. Additionally, the self-originated Al(OH)$_n^-$ is found to adsorb on the anode surface to improve stability. An electrode assembly based on a micropore membrane and CoFeAl layered double hydroxide electrodes operates continuously for 500 hours at 1 A cm$^{-2}$, demonstrating their feasibility in brine electrolysis.

Seawater electrolysis presents a sustainable means of producing hydrogen while conserving freshwater resources and advancing global energy decarbonization[1–3]. However, the presence of Cl$^-$ in seawater can cause the generation of harmful Cl$_2$ or ClO$^-$ on the anode[4,5]. Elevating the alkalinity of the electrolyte is a reliable solution to improve the oxygen evolution reaction (OER) selectivity of anodes, as the standard potential of chlorine oxidation reaction (ClOR) is 0.48 V higher than that of OER (Supplementary Note 1)[6,7]. Moreover, employing 20 wt.% (6 M) NaOH as the electrolyte reduces the solubility of NaCl to less than 3 M due to the common ion effect between NaOH and NaCl (Supplementary Note 2)[8]. This helps to address the issue of ClOR caused by Cl$^-$ accumulation in the brine saturated electrolyte, a consequence of continuous seawater feeding into the electrolyte (Supplementary Fig. 1).

However, the accumulated Cl$^-$ in the brine-containing electrolyte will always lead to OER performance attenuation and anode corrosion. These issues are challenging especially in continuous high-current electrolysis processes. Liu et al. developed RuNiMo electrodes based on the chlorine-repelling effect of MoO$_4^{2-}$, and the as-prepared electrodes operated stably for 300 h under 0.5 A cm$^{-2}$ in an electrolyte containing 2 M NaCl[9]. Li et al. focused on the challenges posed by high-concentration NaCl and developed an electrolytic device that used Na$^+$ exchange membranes to block the migration of Cl$^-$ to the anode side and stably worked at 0.1 A cm$^{-2}$ for 120 h in electrolytes containing 4 M NaCl[10]. Lu et al. synthesized Ag@NiFe-LDH electrodes, and by taking advantage of the strong coordination between Ag$^+$ and Cl$^-$, the electrodes were able to effectively reduce local Cl$^-$ concentration, and functionally operated at 0.4 A cm$^{-2}$ for 450 h in electrolytes containing 2.5 M NaCl[11]. Our research group developed NiFe/NiS$_x$-Ni foam electrodes[12] and CoFePBA/Co$_2$P electrodes[13] through Cl$^-$ repulsion layer design, demonstrating stable operation for 1000 h in electrolytes containing 1.5 M NaCl or higher. Unfortunately, most current reports rely on simulated saline water (only NaCl) as part of the electrolyte, which may not accurately reflect real working conditions. In natural seawater, particularly brine, the presence of complex components, including Br$^-$, can exacerbate electrode corrosion in cooperation with

[1]State Key Laboratory of Chemical Resource Engineering, College of Chemistry, Beijing University of Chemical Technology, Beijing 100029, China. [2]Ocean Hydrogen Energy R&D Center, Research Institute of Tsinghua University in Shenzhen, Shenzhen 518057, China. ✉e-mail: zhoudj@buct.edu.cn; kuangy@tsinghua-sz.org; sunxm@mail.buct.edu.cn

$Cl^-$[14]. Furthermore, the currently reported anodes still lack of stability evaluation at ampere-level current densities and saturated salinity levels. These problems represent a significant obstacle to the industrialization of seawater electrolysis.

In this work, we propose a strategy for highly stable brine electrolysis by utilizing CoFeAl-layered double hydroxide (CoFeAl-LDH) anode materials grown on nickel foam through a facile one-step hydrothermal method. During electrolysis, the amphoteric $Al^{3+}$ dissolves from the LDH layer, giving rise to the formation of defects and $Al(OH)_n^-$. The former promotes activity and the latter protects the anode against corrosion. As a result, the anode can effectively oxidize an electrolyte containing 6-fold concentrated seawater at 2 A cm$^{-2}$ for more than 350 h without corrosion, demonstrating the feasibility of such materials for seawater electrolysis.

## Results

### The challenge of anode corrosion during brine oxidation

Alkaline Seawater electrolysis for hydrogen production is an eco-friendly method (Supplementary Note 1–2)[6–8]. But continuous seawater electrolysis requires the continuous feeding of seawater, resulting in the generation of brine-saturated electrolyte and thus facilitating anode corrosion (Fig. 1a). It is estimated that a seawater electrolysis system with a hydrogen production capacity of 500 Nm$^3$ h$^{-1}$, operating at 10 kA m$^{-2}$ (equal to 1 A cm$^{-2}$), will take just under 3 days to reach electrolyte saturation with NaCl (Supplementary Fig. 1).

Figure 1b shows the in situ electrochemical impedance spectrum (EIS) of commercial Ni foam electrodes at different NaCl concentrations[15,16]. As the $Cl^-$ concentration rises, the phase angle of the electrode decreases at lower potentials, signifying a more pronounced corrosive effect of the brine-containing electrolyte on the electrode (the corrosion mechanism can be seen in Fig. 1a and Supplementary Fig. 2)[17]. Similarly, as shown in Fig. 1c and Supplementary Figs. 3–6, when commercial IrO$_2$ and commercial Ni foam anodes are used in electrolytes containing brine, the accumulation of $Cl^-$ near the anode under the influence of the electric field can cause serious corrosion to the commercial electrode within 3 h, and it is prone to undergo chlorine oxidation side reactions, leading to the formation of toxic substances (Supplementary Fig. 7). These phenomena indicate the urgent need for corrosion-resistant selective oxygen evolution anodes. However, most of the reported anodes for seawater electrolysis lack operational stability under high current and high salinity conditions.

### The activity and stability of CoFeAl-LDH in brine oxidation

The CoFeAl-LDH electrode, prepared via a one-step hydrothermal method, provides a new approach to address these issues. The nanoarray morphology was confirmed by scanning electron microscope (SEM, inset image of Fig. 2b). And the atomic ratio was confirmed by inductively coupled plasma optical emission spectrometer (ICP-OES, Supplementary Table 1) as Co: Fe: Al = 2:1:1. Through careful characterizations including X-ray diffraction, Raman spectrum, X-ray photoelectron spectrum, high-resolution transmission electron microscope, high-angle annular dark field scanning transmission electron microscope, et al (Supplementary Figs. 8–15)[18–25], it was demonstrated that CoFeAl-LDH has been successfully synthesized.

Then a standard three-electrode system was used to check the oxygen evolution reaction (OER) performance of CoFeAl-LDH. Initially, electrolytes containing 1 M NaOH were employed (represented by the dotted line in Fig. 2a), followed by the addition of 0.5 M NaCl (equivalent to the NaCl concentration found in seawater, represented by the solid line in Fig. 2a). The polarization curves clearly indicate that CoFe-LDH experienced a noticeable decline in performance when NaCl was present in the electrolyte, whereas CoFeAl-LDH remained stable. This phenomenon suggests that presence of $Cl^-$ could poison the active sites of CoFe-LDH, but cannot damage the performance of CoFeAl-LDH. Note that the peaks observed at 1.3 V vs RHE in Fig. 2a and 1.22 V vs RHE in Fig. 2b are attributed to the reduction peaks generated by the redox-active Ni and Co elements in the material. And the position of the redox peaks is correlated with the concentration of $OH^-$ in the electrolyte (Supplementary Fig. 36).

Subsequently, an electrolyte consisting of 20 wt.% NaOH, which was saturated (satu.) with NaCl (typically 2.8 M), was used. The CoFeAl-LDH electrode displayed superior catalytic activity compared to both CoFe-LDH (overpotential to achieve 10 mA cm$^{-2}$, $\eta_{10} = 279$ mV) and commercial electrodes (IrO$_2$ and nickel foam), requiring only 256 mV overpotential to achieve 10 mA cm$^{-2}$ (as shown in Fig. 2b and Supplementary Fig. 16). The OER performance and corresponding Tafel slope (Supplementary Fig. 17) of CoFeAl-LDH and CoFe-LDH were recorded in Supplementary Fig. 18, note that these two electrodes exhibited similar activity in low-concentration NaOH, but CoFeAl-LDH exhibited significantly better performance ($\Delta\eta_{10} = \eta_{10, \text{CoFe-LDH}} - \eta_{10, \text{CoFeAl-LDH}} = 23$ mV) in high-concentration NaOH. According to previous investigations made by our group and other reports[26–28], this improvement was attributed to the dissolution of the amphoteric metal $Al^{3+}$ in the CoFeAl-LDH electrode, which induced formation of $M^{3+}$ ion defects and exposure of more active sites.

Long-term stability is a crucial factor for the feasibility of electrodes in practical seawater electrolysis. Therefore, we conducted tests on the chronopotentiometry (CP) response of the two-electrode system, using CoFe-LDH or CoFeAl-LDH as the anode, commercial nickel foam as the cathode and 20 wt.% NaOH + satu. NaCl solution as the electrolyte. The test was conducted under industrially current densities ranging from 0.2 to 1.0 A cm$^{-2}$ (Fig. 2c), simulating the concentrated brine electrolysis process. The results showed that the CoFe-LDH electrode was partially corroded within 50 h (Supplementary Fig. 19). In contrast, the CoFeAl-LDH electrode remained stable at

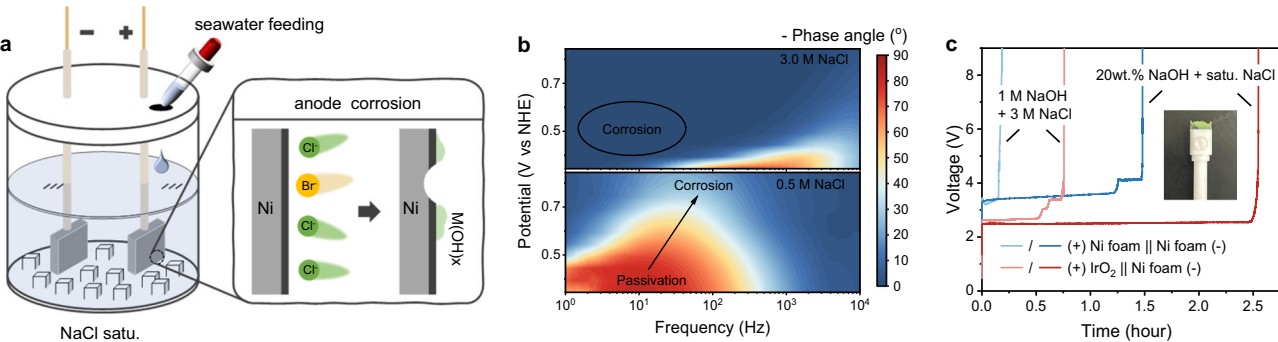

**Fig. 1 | Anode corrosion problem faced by brine oxidation. a** The scheme of continuous seawater feeding during seawater electrolysis for hydrogen production, and the corrosion mechanism faced by the anodes. **b** The in situ EIS of commercial Ni foam electrode at different NaCl concentrations. **c** Chronopotentiometry (CP) response of commercial Ni foam and commercial IrO$_2$ electrodes as the anodes at 0.2 A cm$^{-2}$, respectively.

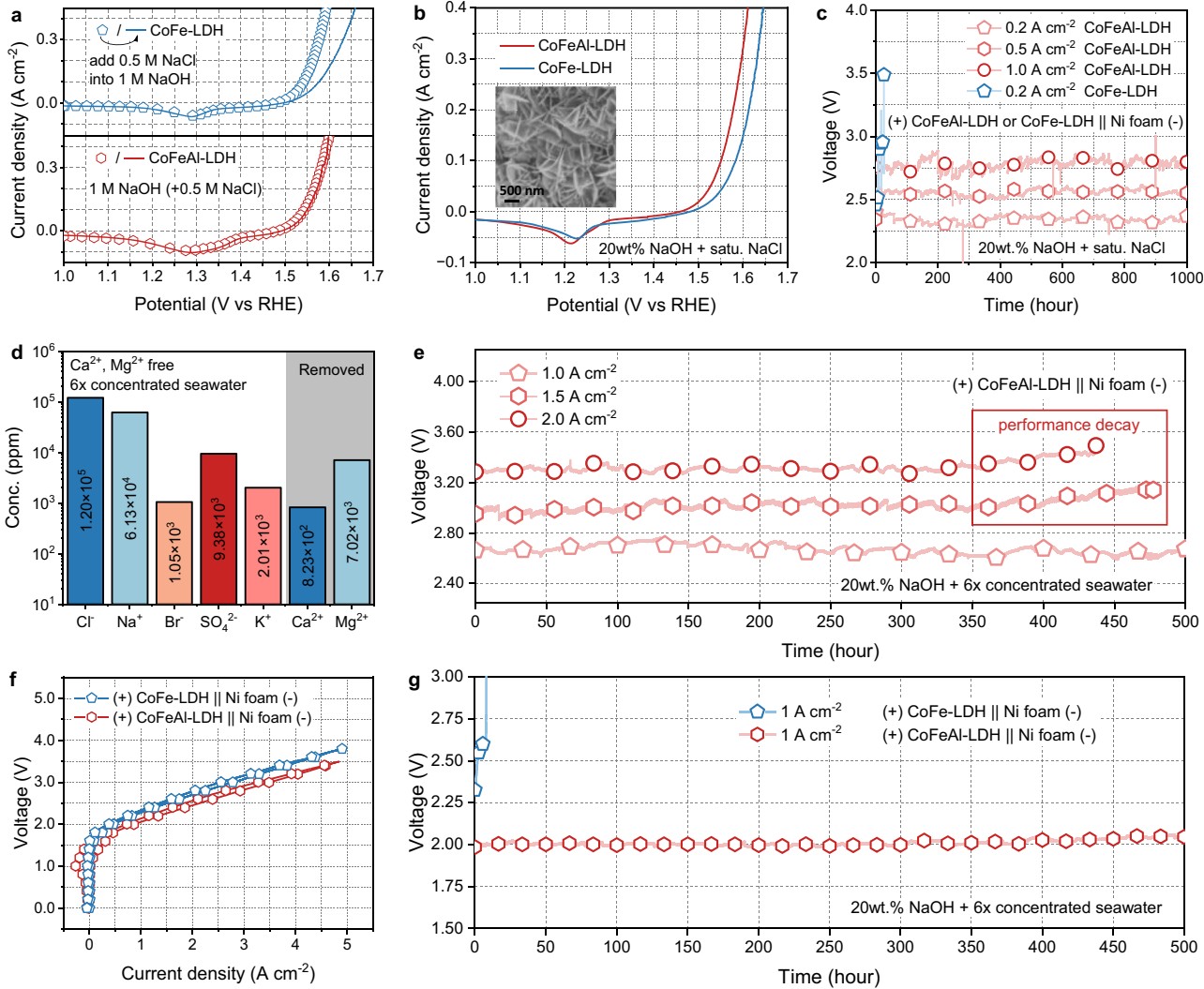

**Fig. 2 | Electrocatalytic brine oxidation performance of CoFeAl-LDH.** Polarization curves of CoFe-LDH and CoFeAl-LDH in (**a**) 1 M NaOH (dotted line), 1 M NaOH + 0.5 M NaCl (solid line), and (**b**) 20 wt.% NaOH + saturated (satu.) NaCl. Inset is SEM image of CoFeAl-LDH. **c** CP response at 0.2, 0.5 and 1.0 A cm$^{-2}$ in an electrolyte containing 20 wt.% NaOH + satu. NaCl. **d** The concentration (conc.) of ions in 6-fold concentrated seawater. **e** CP response at 1.0, 1.5 and 2 A cm$^{-2}$ in electrolyte containing 20 wt.% NaOH + 6-fold concentrated seawater, respectively. **f** Polarization curves and (**g**) CP response of membrane electrode assemblies (MEA).

current densities tested from 0.2 to 1 A cm$^{-2}$. Furthermore, the polarization curves (Supplementary Fig. 20) demonstrated that even after 1000 h of stability testing, the OER performance of the CoFeAl-LDH electrode did not significantly decay (the decay ratio is less than 1.2%). This indicates that the CoFeAl-LDH electrode exhibits resistance to Cl$^-$ corrosion efficiently.

To further demonstrate the performance of CoFeAl-LDH electrodes in brine electrolysis, a more rigorous testing environment was implemented. Real seawater (from the Yellow Sea in China) was concentrated by a factor of 6-fold. The components of the brine were analyzed using ion chromatography and ICP-OES (Fig. 2d and Supplementary Table 5). The analysis revealed that, in addition to containing ultra-high concentrations of Cl$^-$ (119.86 g L$^{-1}$, equivalent to 1.20 × 10$^5$ ppm), the brine also had higher levels of corrosive Br$^-$ (1.05 × 10$^3$ ppm) and other ions (SO$_4^{2-}$, K$^+$, Na$^+$, etc.), which increased the difficulty of anti-corrosion during the electrolysis process[14]. After removing Ca$^{2+}$ and Mg$^{2+}$ and adding 20 wt.% NaOH, the brines were used as electrolytes. And the OER activity of CoFeAl-LDH in this electrolyte was first confirmed to be significantly unaffected (Supplementary Fig. 34). Figure 2e illustrates the CP response of the (+) CoFeAl-LDH || Ni foam (−) configuration in electrolytes containing 20 wt.% NaOH mixed with

6-fold concentrated seawater at current densities of 1.0, 1.5, and 2.0 A cm$^{-2}$, respectively. Throughout the 500 h stability testing at 1 A cm$^{-2}$, the increase in voltage is approximately 0.23%, and the CoFeAl-LDH anode evolved only O$_2$ exclusively (Supplementary Figs. 21–22). Even by increasing the current density to 1.5 or 2.0 A cm$^{-2}$, the anode can operate steadily for over 350 h, suggesting the feasibility of the anode working under fluctuating power conditions. The CP tests in Supplementary Fig. 23 determined the maximum Cl$^-$ concentration that the CoFeAl-LDH electrode can withstand, it can be observed that the CoFeAl-LDH electrode demonstrated higher Cl$^-$ tolerance than CoFe-LDH and can withstand a Cl$^-$/OH$^-$ ratio of up to 4/1 (4 M NaCl/1 M NaOH). These remarkable stabilities made CoFeAl-LDH outstanding from previous electrodes[9,11,13,29–33] and highlight its potential for practical applications in the continuous seawater electrolysis (Supplementary Fig. 24 and Supplementary Table 2).

Finally, a membrane electrode assembly (MEA, Supplementary Fig. 25) based on microporous membrane (Zirfon UPT 500, Agfa-Gevaert N.V.) was assembled and evaluated. Figure 2f and Supplementary Fig. 26 illustrate that the MEA equipped with CoFeAl-LDH as the anode displayed remarkable electrocatalytic activity, requiring only 2.06 V to achieve a current density of 1.0 A cm$^{-2}$, and the

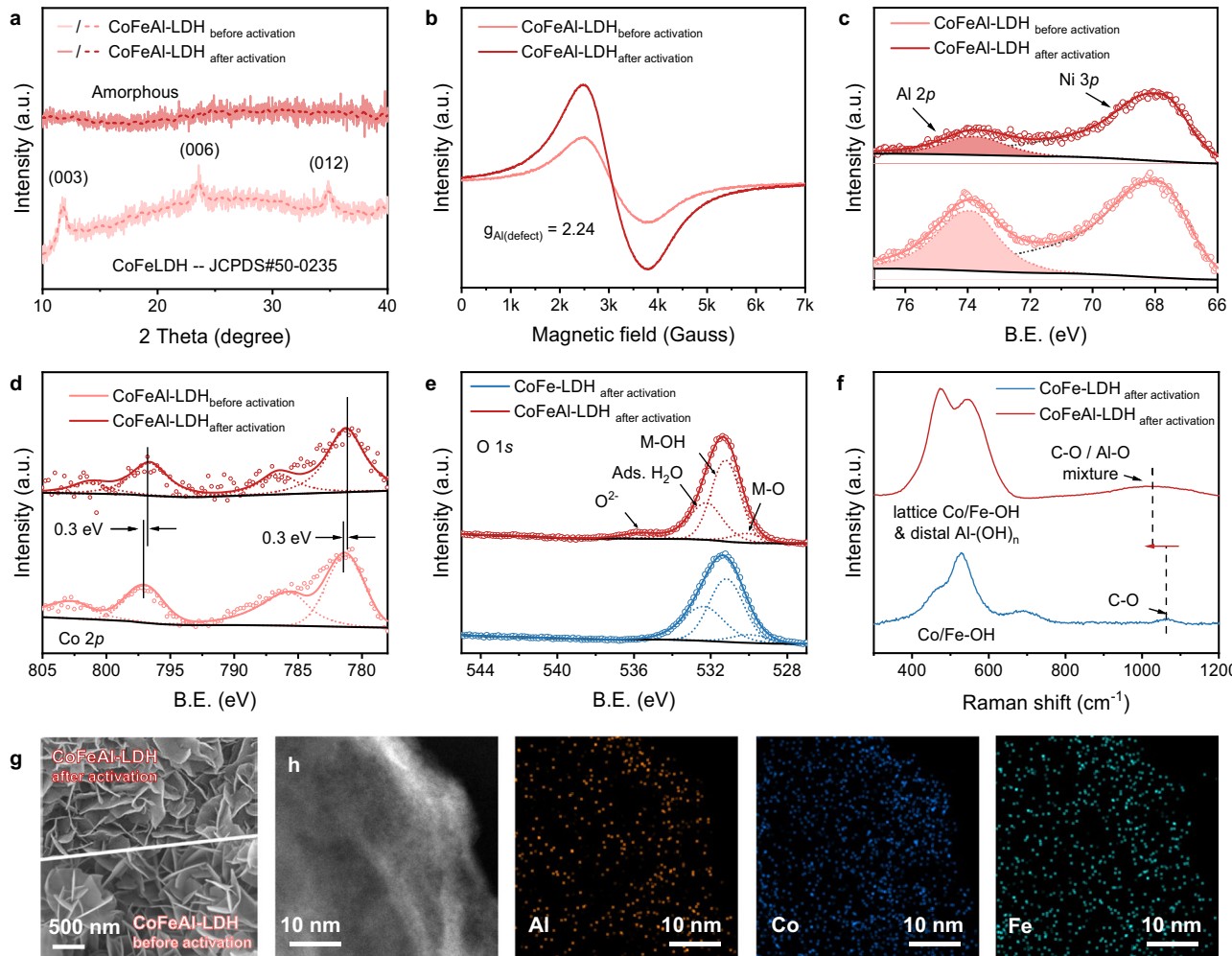

**Fig. 3 | Structure characterizations of CoFeAl-LDH after electrocatalytic brine oxidation. a** XRD patterns, (**b**) EPR spectrum and (**c**) Al 2*p* and (**d**) Co 2*p* high-resolution XPS spectrum of CoFeAl-LDH before and after activation. Note that B.E. is an abbreviation for binding energy, and the units of intensity are arbitrary units (a.u.). **e** O 1*s* high-resolution XPS spectrum and (**f**) Raman spectrum of CoFe-LDH and CoFeAl-LDH after activation, ads. means adsorbed. **g** SEM images of CoFeAl-LDH before and after activation. **h** HAADF-STEM image of CoFeAl-LDH after activation and the corresponding EDS elemental mapping for Co, Fe and Al.

corresponding energy consumption (E.C.) is only 4.93 kWh m$^{-3}$ H$_2$. The overall system exhibited a low contact resistance of 0.11 Ω cm$^2$ (Supplementary Fig. 27). Additionally, the stability of the MEA was examined, and as depicted in Fig. 2g, it demonstrated the ability to operate consistently for over 500 h at 1.0 A cm$^{-2}$, with a mere 1.69% voltage decay. This outcome signifies the practical applicability of the CoFeAl-LDH electrode in brine electrolysis, maintaining stability even at extremely high current densities.

**Structure characterizations of reconstructed CoFeAl-LDH**
To investigate the origin of the remarkable performance, characterizations were performed on the CoFeAl-LDH before and after activation. It is worth noting that the activation process involved a 10-h CP test in the 20 wt.% NaOH + satu. NaCl electrolyte at 0.2 A cm$^{-2}$. As shown in Fig. 3a, the X-ray diffraction (XRD) pattern reveals that following activation, CoFeAl-LDH turned out to be amorphous, suggesting the reconstruction during the OER process. Additionally, as shown in the high-angle annular dark field scanning transmission electron microscope (HAADF-STEM) images (Supplementary Fig. 35), the activated CoFeAl-LDH exhibits a reduced crystallinity, further corroborating the occurrence of reconstruction. Then the local chemical environments of CoFeAl-LDH were confirmed by electron paramagnetic resonance (EPR, Fig. 3b) spectrum. The much stronger signal visible at g = 2.24 can be attributed to the signal of electrons trapped

by Al$^{3+}$ vacancies in CoFeAl-LDH after dissolving Al$^{3+}$[26,34]. Furthermore, the weaker signal of high-resolution Al 2*p* X-ray photoelectron spectrum (XPS) after activation further supports the generation of Al$^{3+}$ vacancies (Fig. 3c). The findings lend credence to the notion that the enhancement in OER performance of CoFeAl-LDH, following an upsurge in NaOH concentration, can be attributed to the generation of Al$^{3+}$ vacancies, as elaborated in Fig. 2b[26,28]. Meanwhile, the high-resolution Co 2*p* XPS of both CoFeAl-LDH before and after activation (Fig. 3d) shows a pair of peaks assigned to Co$^{2+}$ 2*p*$_{3/2}$ and 2*p*$_{1/2}$[35-37]. However, the signal shift towards lower binding energy (B.E.) can be observed after activation, similar to the CoFe-LDH in Supplementary Fig. 10, indicating that the electron cloud density near Co decreases due to the influence of Al$^{3+}$ dissolution and oxidation potential. Moreover, high-resolution O 1*s* XPS spectrum was obtained for the CoFe-LDH and CoFeAl-LDH electrodes (Fig. 3f). Apart from the signals corresponding to M-O, M-OH, and adsorbed H$_2$O located at 530.1, 531.3, and 532.3 eV, respectively, CoFeAl-LDH exhibits an additional signal at 535.9 eV, which corresponds to the O$^{2-}$ ions in the oxygen-deficient regions through etching[38,39].

Subsequently, Raman spectrum was obtained on the activated CoFe-LDH and CoFeAl-LDH electrodes (Fig. 3e). Comparing with Supplementary Fig. 9, it is evident that the bending vibration peak of M-O in CoFe-LDH remains relatively unchanged. In contrast, CoFeAl-LDH exhibits a completely different Raman spectrum. The peaks observed

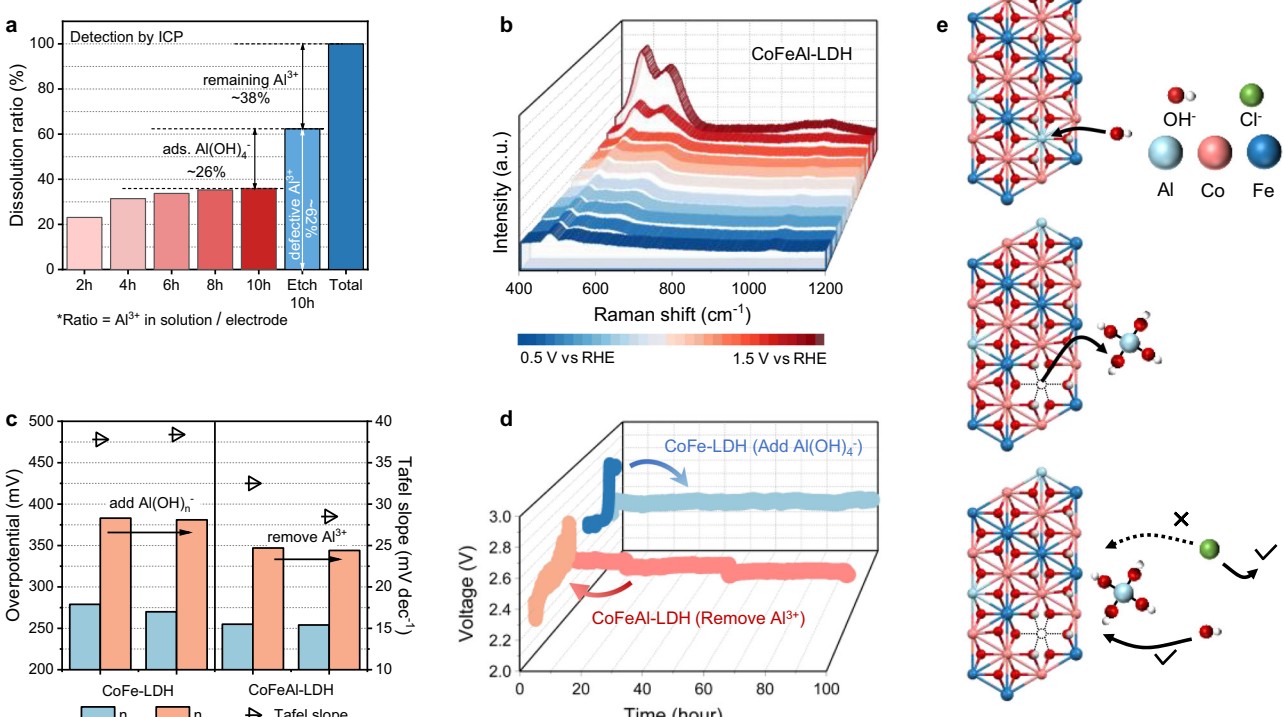

**Fig. 4 | Investigation on the origin of corrosion resistance during brine oxidation. a** Dissolution ratio of Al in CoFeAl-LDH. **b** Operando Raman spectrum of CoFeAl-LDH, potential range: OCP-1.5 V vs RHE. **c** OER performance and (**d**) CP response at 0.2 A cm$^{-2}$ of CoFe-LDH before/after adding Al(OH)$_n^-$ to the electrolyte, and CoFeAl-LDH before/after etching Al$^{3+}$, in an electrolyte containing 20 wt.% NaOH + satu. NaCl. **e** Schematic diagram of Al$^{3+}$ dissolution and Al(OH)$_n^-$ generation.

between 400–700 cm$^{-1}$ are attributed to the lattice M-OH bending vibrations, which are combined with the bending vibrations of Al(OH)$_n^-$ species formed during the etching process. Notably, the broad peak at 1050 cm$^{-1}$ displays a more pronounced red shift compared to the C–O signal (1073 cm$^{-1}$) in Supplementary Fig. 9. This shift is attributed to the increased stretching vibrations of Al–O bonds from Al(OH)$_n^-$ species[40,41].

Scanning electron microscope (SEM) images in Fig. 3g show the similar two-dimensional nanoarray morphologies of the CoFeAl-LDH before and after activation. These structures have long been known to exhibit superaerophobicity in electrolytic solutions, which facilitates the mass transport of O$_2$ products[23,24]. In addition, the LDH nanosheets became more wrinkled after activation, consistent with prior findings indicating that the Al$^{3+}$ within the material was etched. The HAADF-STEM image in Fig. 3h further supports this conclusion, and corresponding energy-dispersive X-ray spectroscopy mapping (EDS-mapping) analysis confirms a reduction in Al content (in comparison to Supplementary Fig. 14). Furthermore, the distribution of Co, Fe, and Al remains uniform within the LDH plate, suggesting that there was no phase separation. This is advantageous for preserving the electrocatalytic OER performance[42].

## The self-protective properties of CoFeAl-LDH

Real-time monitoring of the electrode's condition during the OER process is also crucial to gain a deeper understanding of performance. The ratio of Al$^{3+}$ etched during the activation process and the total Al content in the electrode were detected using ICP-mass spectrometry (ICP-MS), respectively. As shown in Fig. 4a and Supplementary Table 3, when the activation time reached 8 h, approximately 36% of the Al$^{3+}$ was etched and reached equilibrium. In contrast, when the CoFeAl-LDH electrode was immersed in a 20 wt.% NaOH + satu. NaCl solution for 10 h, the dissolved content of Al$^{3+}$ could reach 62%. This implies that

applying an oxidation potential to the CoFeAl-LDH during Al etching facilitates the adsorption of resultant Al(OH)$_n^-$ on the electrode surface, with the adsorption ratio accounting for approximately 26% of the total Al$^{3+}$ in the electrode. Moreover, it is important to note that not all Al$^{3+}$ in the electrode dissolves. At least approximately 38% of the Al$^{3+}$ remains within the LDH even after long-term etching at highly concentrated NaOH.

Figure 4b and Supplementary Fig. 28 present the operando Raman spectrum of CoFeAl-LDH and CoFe-LDH electrodes over the potential range from open circuit potential (OCP) to 1.5 V vs RHE. It is evident that when the applied potential reaches 1.4 V vs RHE, the peaks within the 400–700 cm$^{-1}$ range become more pronounced compared to CoFe-LDH. As previously mentioned, this can be attributed to the etched Al$^{3+}$ adsorbing at the distal sites of electrodes and coordinating greater amounts of OH$^-$ from the electrolyte to generate Al(OH)$_n^{-[40,41]}$. Besides, at the potential range of 1.3 to 1.5 V vs RHE, the peak of Al–O stretching vibration centered at 988 cm$^{-1}$ in Fig. 4b exhibits a considerable enhancement in signal intensity, further supporting this observation[21]. Furthermore, the applied potential on the CoFeAl-LDH electrode was gradually decreased from 1.5 V vs RHE to OCP (Supplementary Fig. 29). Interestingly, it was observed that the peaks within the 400–700 cm$^{-1}$ wavenumber range did not exhibit any significant change, and the Al–O stretching vibration signal at 988 cm$^{-1}$ can still be observed until the potential is reduced to OCP. This result indicates the highly stable adsorption of Al(OH)$_n^-$ species on the electrode surface, as illustrated in the schematic diagram in Fig. 4e.

To verify whether the enhanced corrosion resistance of the electrode is due to the adsorbed Al(OH)$_n^-$, 7.8 ppm Al(OH)$_n^-$, which corresponds to the amount of Al$^{3+}$ dissolution during immersion, was added to the electrolyte when using CoFe-LDH working electrode. Furthermore, CoFeAl-LDH after Al-removal was used as a reference sample. Figure 4c and Supplementary Fig. 28 depict the overpotential

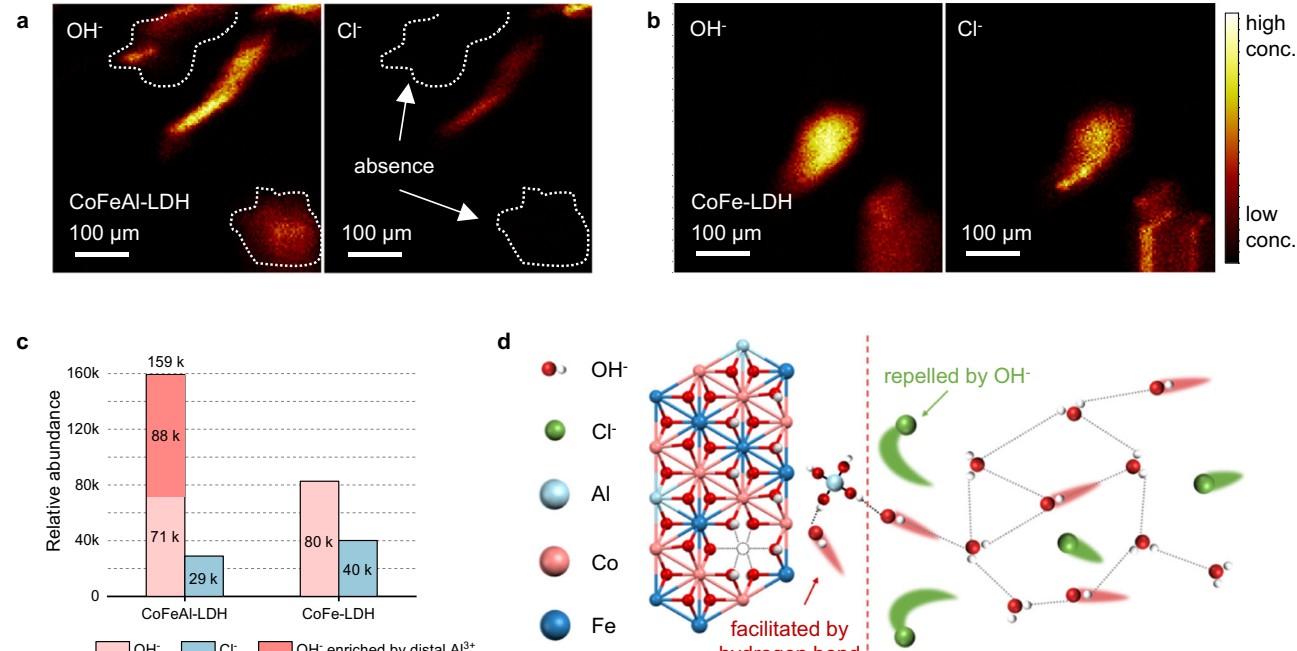

**Fig. 5 | Investigation on the role of self-originated Al(OH)₄⁻ in facilitating OH⁻ enrichment.** TOF-SIMS mapping of OH⁻ and Cl⁻ fragments from (**a**) CoFeAl-LDH electrode and (**b**) CoFe-LDH electrode surface after activation in 20 wt.% NaOH + satu. NaCl. **c** TOF-SIMS relative abundance of OH⁻ and Cl⁻ from CoFeAl-LDH electrode and CoFe-LDH electrode surface. **d** Illustration of the corrosion resistance mechanism of surface adsorbed Al(OH)$_n^-$.

and Tafel slope of these samples before and after the treatment. It can be observed that the activity of both the CoFe-LDH and CoFeAl-LDH electrodes remained almost unchanged before and after the treatment. However, as shown in Fig. 4d and Supplementary Fig. 31, the stability of CoFe-LDH was significantly improved after the addition of Al(OH)$_n^-$ to the electrolyte, with the performance degradation being less than 0.29% after 500 h of stability test at 0.2 A cm⁻². But it reaches 16.41% in only 10 h in the absence of Al(OH)$_n^-$. Moreover, the stability of CoFeAl-LDH decreased significantly after the removal of Al³⁺. These results suggest that the adsorption of Al(OH)$_n^-$ on the electrode surface is crucial for achieving high stability.

## Self-originated Al(OH)$_n^-$ enriched OH⁻

To reveal the state and role of Al(OH)$_n^-$, time-of-flight secondary ion mass spectrometry (TOF-SIMS) was used to detect the concentrations of OH⁻ and Cl⁻ on the surface of the activated electrode. The TOF-SIMS mapping in Fig. 5a reveals that the CoFeAl-LDH surface is covered with OH⁻, while the signal of Cl⁻ is lower than the detection limit at the same dashed area. Only in the area with a very high OH⁻ signal, the weak Cl⁻ signal can be observed. Conversely, in the TOF-SIMS mapping of CoFe-LDH (Fig. 5b), the locations of these two ions are consistent, indicating that the CoFe-LDH cannot repel Cl⁻ whilst CoFeAl-LDH electrode has a strong tendency to repel Cl⁻. Additionally, the distribution of Al(OH)$_n^-$ species on the CoFeAl-LDH surface is depicted in Supplementary Fig. 32, and their relative abundances are presented in Supplementary Table 4. The results in Fig. 5c further support the above findings, showing that the relative intensity of OH⁻ adsorbed on the surface of CoFe-LDH electrodes is 80 k, while Cl⁻ is 40 k. This ratio is approximately the same as the concentration ratio of OH⁻ and Cl⁻ in the electrolyte (OH:Cl ≈ 2:1), proving that CoFe-LDH does not exhibit selectivity for the adsorption of OH⁻ or Cl⁻. However, when CoFeAl-LDH is utilized, the absorbed Al³⁺ enriched OH⁻ species, resulting in a surface OH⁻ concentration that is 5-fold as high as the Cl⁻ concentration.

Figure 5d presents a schematic diagram of the CoFeAl-LDH electrode which repels Cl⁻ but admits OH⁻ simultaneously. First of all, OH⁻ typically has a higher migration rate in aqueous solution than Cl⁻[43]. As

shown in Supplementary Fig. 33, OH⁻ transmission in the aqueous electrolyte relies on the hydrogen bond network formed between adjacent water molecules, typically referred to as Grotthuss transfer mechanism[44–46]. This mechanism ensures that OH⁻ transmission can efficiently reach hydroxide ion surrounded Al(OH)$_n^-$. Thus, OH⁻ adsorption is selectively enhanced near to the electrode surface with Al(OH)$_n^-$ coverage[47,48]. More importantly, due to the presence of negatively charged Al(OH)$_n^-$ species on the electrode surface with enriched OH⁻, strong repelling Coulombic forces are generated by OH⁻ against Cl⁻, as indexed by the dashed line in Fig. 5d[42,49,50]. This design simultaneously improves the selectivity of the OER while preventing the adsorption of Cl⁻ and avoiding corrosion of the electrode.

## Discussion

CoFeAl-LDH anodes demonstrated their capability for efficient and stable oxidation in concentrated brine at 1.0–2.0 A cm⁻² current densities for more than 350 h. A novel mechanism for the dual enhancement of activity and stability was proposed: etching by OH⁻ during OER process leads to the self-origination of Al³⁺ vacancies in CoFeAl-LDH that boost OER activity. Simultaneously, the formation of Al(OH)$_n^-$ species partially adsorbed onto the electrode surface to enrich OH⁻ and repel Cl⁻ through Coulombic forces, thereby facilitating the transport of other OH⁻ to the electrode surface via hydrogen bonding. These materials are facile to large scale production and can be used without complicated processing, becoming activated in situ, thus benefiting industrial assembly and operation. This selective enhancement of OH⁻ adsorption mechanism should shed light on further electrode optimization to prevent corrosion caused by Cl⁻ in continuous brine electrolysis.

## Methods
### Materials
Co(NO₃)₂·6H₂O, Fe(NO₃)₃·9H₂O, Al(NO₃)₃·9H₂O were purchased from Aladdin Industrial Co. Urea, NaOH, NaCl were purchased from Beijing Chemical Reagents Co. Deionized water with a resistivity ≥ 18 MΩ was used to prepare all aqueous solutions. All of the reagents were of analytical grade and were used directly without further purification.

## Synthesis of CoFeAl·LDH electrode

All chemicals were used as received without further purification. First, 1 mmol $Co(NO_3)_2 \cdot 6H_2O$, 0.5 mmol $Fe(NO_3)_3 \cdot 9H_2O$, 0.5 mmol $Al(NO_3)_3 \cdot 9H_2O$ and 5 mmol urea were dissolved in 30 mL deionized water. Then this solution and a piece of $2*4\ cm^2$ cleaned Ni foam was transferred into the Teflon-lined stainless autoclave and maintained at 120 °C for 12 h to get the CoFeAl·LDH electrode.

## Synthesis of CoFe·LDH electrode

All chemicals were used as received without further purification. First, 1 mmol $Co(NO_3)_2 \cdot 6H_2O$, 1 mmol $Fe(NO_3)_3 \cdot 9H_2O$ and 5 mmol urea were dissolved in 30 mL deionized water. Then this solution and a piece of $2*4\ cm^2$ cleaned Ni foam was transferred into the Teflon-lined autoclave and maintained at 120 °C for 12 h to get the CoFe-LDH electrode.

## Physical characterization

SEM images were obtained on a Zeiss SUPRA55 scanning electron microscope, which was operated at 10 kV. HRTEM images were carried out by the JEOL JEM-2100 operating at 200 kV. HAADF-STEM images were obtained using the FEI-Titan G2 at 300 kV, and the corresponding EDS elemental mappings were characterized using an energy-dispersive spectrometer (Oxford). XRD patterns were recorded on an Ultima IV (Rigaku) in the range from 10° to 40° at a scan rate of 1° $min^{-1}$. XPS were performed by using a model of K-Alpha (Thermo Scientific). Raman spectra were recorded on a Lab RAM Aramis (HORIBA Jobin Yvon S.A.S, Laser 532). EPR were characterized using Bruker EMX PLUS at 9.30 GHz and 100 K. ICP-OES and ICP-MS were carried out by the Thermo Fisher iCAP 7400. Ion chromatography was carried out by a 905 Titrando (Metrohm). And TOF-SIMS Mapping were obtained on an IONTOF 5.

## Electrochemical measurements

The electrochemical measurements for OER were performed in a standard three-electrode system at room temperature (25 °C) on an electrochemical workstation (CHI 660D, Chenhua, Shanghai), where CoFeAl·LDH and CoFe-LDH serve as the working electrodes, while Pt foil electrode and SCE electrode serve as counter electrode and reference electrode, 1 M NaOH + 0.5 M NaCl and 20wt.% NaOH + satu. NaCl were used as the electrolyte, respectively. The LSV tests at a scan rate of $2\ mV\ s^{-1}$ were performed after 40 cycles of CV. The final potentials were converted with respect to the reversible hydrogen electrode (RHE) based Nernst equation:

$$E\,(\text{RHE}, iR\ \text{corrected}) = E(\text{SCE}) + 0.244\,\text{V} + 0.059\,\text{pH} - iR \qquad (1)$$

where $i$ is the measured current density and $R$ is the solution resistance determined by electrochemical impedance spectroscopy at a high frequency. The Tafel plots were derived from polarization curves at low overpotentials fitted to the Tafel equation:

$$\eta = a + b\,\log J \qquad (2)$$

where $\eta$ is the overpotential, $J$ is the current density, a and b are constants. The CP tests were carried out at current densities of 0.2, 0.5, 1.0, 1.5, and $2.0\ A\ cm^{-2}$ in a two-electrode system using 20wt.% NaOH + satu. NaCl or 20wt.% NaOH + concentrated real seawater electrolyte with a piece of Ni foam as a cathode and CoFeAl·LDH or CoFe-LDH as an anode. The obtained data for CP tests for stability were not $iR$ corrected and displayed as raw data.

## MEA test

The MEA was based on the microporous membrane (Zirfon UPT 500, Agfa-Gevaert N.V.), and the catalyst-coated substrate (CCS) method was used. The CoFeAl·LDH or CoFe-LDH electrode was used as the anode, the Ni foam served as the cathodes. Finally, the cell was integrated by pressing the anode, microporous membrane, cathode and two Ti end plates with flow field channel. The active area was regarded as $4\ cm^2$, which was the area covered by the serpentine flow channel. During the test, the cell was maintained at 80 °C, and the pre-heated electrolyte was fed to both the anode and cathode at a flow rate of $50\ mL\ min^{-1}$. The stability of the MEA was measured at $10\ kA\ m^{-2}$. All the data of were not $iR$ corrected and displayed as raw data.

## Operando electrochemical Raman measurements

Electrochemical Raman spectroscopy was carried out on a Horiba Lab RAM HR Evolution confocal Raman spectrometer with a 532-nm laser source, a 50× objective and an acquisition time of 90 s. And the electrochemical workstation (CHI 660D, Chenhua, Shanghai) were used to performed the Amperometric i-t Test under OCP-1.5 V vs RHE, respectively.

## Data availability

The data generated in this study are provided in the Source Data file. Source data are provided with this paper.

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

## Acknowledgements

This work was financially supported by the National Key Research and Development Program of China (No. 2021YFA1502200, received by Y.K.), the National Natural Science Foundation of China (No. 21935001, received by X.S.), the National Key Beijing Natural Science Foundation (No. Z210016, received by X.S.), and the long-term subsidy mechanism from the Ministry of Finance and the Ministry of Education of China.

## Author contributions

X.S., Y.K., D.Z., and W.L. designed this project and the experiment. W.L. conducted the majority of the experiments. T.L. and S.L. performed Raman tests, B.D. and Q.S. carried out the XPS tests. A.C. and W.L. draw the schematic diagrams. X.G. and W.L. conducted the MEA tests. X.S., Y.K., D.Z., J.Y., and W.L. co-wrote the paper. The results and manuscript are commented and discussed by all authors.

## Competing interests

The authors declare no competing interests.
