## [Peer Review File · Nature Communications]

REVIEWER COMMENTS

Reviewer #1 (Remarks to the Author):

Hydrogen production through seawater electrolysis is a promising and attractive avenue for sustainable energy utilization. However, the generation of concentrated brine during electrolysis is inevitable, which makes it necessary for the anode have complex preparation processes to achieve anti-corrosion properties. In practical applications, this unavoidably raises the cost and difficulty of electrode preparation.

In this manuscript, the authors have simplified the electrode preparation process, and proposed a one-step synthesis method for CoFeAl-LDH electrodes. They demonstrated that Al^{3+} in the electrodes is etched during activation, creating M^{3+} vacancies and generating $Al(OH)_n^-$. Thus achieving a dual improvement in OER performance and corrosion resistance.

Moreover, unlike previous studies, the authors conducted long-term electrolysis using concentrated brine obtained by evaporation of real seawater, providing confidence in the practical application of seawater electrolysis.

It is recommended to be publication in Nature Communications after minor revision. The detailed comments are as follows:

1. In Figure 2e, when the electrolyte contains concentrated brine, the performance decay occurs after 350 hours with a current density of 1.5 A/cm² or higher. I encourage the authors analyze whether this phenomenon is related to electrode corrosion.
2. Some studies have shown that the use of real seawater may lead to a decay in anode activity. This issue may be even more severe in concentrated seawater. The authors are requested to verify whether the OER performance of CoFeAl-LDH is affected in the electrolyte containing brine.
3. For the first time, the authors have proposed hydrogen bonding network to repelled Cl^- , please give more discussion on this mechanism and analyze whether there are other factors that may influence the corrosion resistance ability.

Reviewer #2 (Remarks to the Author):

Compared to the maturity of desalination technology, direct seawater electrolysis is often criticized for its high cost and continuous operation challenges. However, this work not only demonstrates the feasibility of low-cost, high-stability direct seawater electrolysis, but also proposes a possible solution to the saline water concentration problem caused by seawater desalination. In addition to promoting academic research on seawater electrolysis, this work has the potential to significantly propel industrialization in this field. To further improve this manuscript, I have the following suggestions.

1. It is noteworthy that the brine employed by the authors is 6X concentrated seawater. Please supply the composition analysis of the seawater before concentration and explain the possible impact of various substances present on seawater electrolysis.
2. In industry, alkaline water electrolyzers typically operate at current densities below 1 A cm⁻². And the MEA used by the authors can stably operate at 1 A cm⁻² even when using 6X concentrated seawater as the electrolyte, which is encouraging. I think it would be valuable if the authors could do a costing account based on this electrolyzer.
3. Chlor-alkali electrolyzers are also widely used for brine utilization, enabling simultaneous production of H_2 , Cl_2 , and NaOH. A comparative analysis needs to be conducted to assess their respective advantages and disadvantages.
4. In summary, the progress achieved in this work is important in the field of seawater electrolysis, and I highly recommend its publication in Nature Communications.

Reviewer #3 (Remarks to the Author):

This study developed an CoFeAl layered double hydroxide (CoFeAl-LDH) anodes, which shows excellent oxygen evolution reaction (OER) activity and stability. The Al^{3+} of CoFeAl-LDH in electrode were etched off by OH^- during OER process, resulting in M^{3+} vacancies that boost OER activity. Additionally, the self-originated $Al(OH)_n^-$ was found adsorb on the anode surface to improve stability. However, before this paper could be considered for publication, some main issues should be well addressed.

(1) The authors should describe the peaks that appear in the LSV curves of Figures 2a and 2b and explain why the peaks in Figures 2a and 2b are not in the same position.

(2) In Figure 3g, the after-activation CoFeAl-LDH still maintains a sheet-like morphology of LDH, but the XRD of the after-activation CoFeAl-LDH in Figure 3a no longer shows peaks corresponding to LDH sheets. The author should provide an explanation for this discrepancy.

(3) In Figure 3d, regarding the XPS analysis of Co 2p, the peak area ratio of Co 2p_{3/2} to Co 2p_{1/2} for the same oxidation state should be 2:1, and the full width at half maximum (FWHM) should be consistent. However, it does not meet these requirements. The author should review the XPS peak decomposition rules for this element and make the necessary adjustments.

(4) In line 51, there is a numerical omission in "evaluation at $A\ cm^{-2}$ ". The author should make the correction. Similar language errors can be observed elsewhere, so the author should pay attention to writing standards.

(5) Between lines 73-74, the correct conversion for $1\ A\ cm^{-2}$ should be $10000\ A\ m^{-2}$, while the text states $1000\ A\ m^{-2}$. The author should make the necessary correction and ensure thorough attention to detail in the writing to avoid such errors.

(6) The value of the overpotential in line 123 of the main text is incorrectly written.

(7) In Supplementary Figure 16, why are the current densities corresponding to each sample at a potential of 1.0 V not concentrated at the same point? The authors are requested to explain.

(8) The OH in line 294 of the text lacks the superscript "-".

(9) The subscript "n" is missing from $Al(OH)_n^-$ in line 300 of the text.

Response to Reviewers

Reply to Reviewer 1

Hydrogen production through seawater electrolysis is a promising and attractive avenue for sustainable energy utilization. However, the generation of concentrated brine during electrolysis is inevitable, which makes it necessary for the anode have complex preparation processes to achieve anti-corrosion properties. In practical applications, this unavoidably raises the cost and difficulty of electrode preparation.

In this manuscript, the authors have simplified the electrode preparation process, and proposed a one-step synthesis method for CoFeAl-LDH electrodes. They demonstrated that Al^{3+} in the electrodes is etched during activation, creating M^{3+} vacancies and generating $Al(OH)_n$. Thus achieving a dual improvement in OER performance and corrosion resistance.

Moreover, unlike previous studies, the authors conducted long-term electrolysis using concentrated brine obtained by evaporation of real seawater, providing confidence in the practical application of seawater electrolysis.

It is recommended to be publication in Nature Communications after minor revision. The detailed comments are as follows:

Response:

We appreciate the reviewer's insightful and helpful comments on our manuscript. The reviewer's suggestions and criticisms help us substantially improve the quality of the manuscript. We have addressed the comments point-by-point as follows.

- 1. In Figure 2e, when the electrolyte contains concentrated brine, the performance decay occurs after 350 hours with a current density of $1.5 A/cm^2$ or higher. I encourage the authors analyze whether this phenomenon is related to electrode corrosion.*

Response:

Thank you very much for your valuable comment. The electrodes are highly susceptible to fracture after corrosion (Supplementary Figure 19). However, as depicted in Appendix Figure 1a, the CoFeAl-LDH electrode maintains its structural integrity despite performance degradation.

Furthermore, upon corrosion initiation, the electrode usually experiences a rapid rate of performance decay, as evidenced by CoFe-LDH in Figure 2d and several samples in Figure 1b. In contrast, when operated at current densities exceeding 1.5 A cm^{-2} , the performance degradation of CoFeAl-LDH is relatively slow.

Simultaneously, SEM images of corroded CoFe-LDH and performance-degraded CoFeAl-LDH (Appendix Figure 1b, c) reveal the presence of numerous cracks on the former, while the latter preserves its nanoarray structure. These observations suggest that the performance decay of CoFeAl-LDH cannot be solely attributed to the occurrence of corrosion.

Appendix Figure 1. (a) The photograph of performance-degraded CoFeAl-LDH. SEM images of (b) corroded CoFe-LDH and (c) performance-degraded CoFeAl-LDH.

Supplementary Figure 19. Photograph of CoFe-LDH electrode after being corroded.

2. *Some studies have shown that the use of real seawater may lead to a decay in anode activity. This*

issue may be even more severe in concentrated seawater. The authors are requested to verify whether the OER performance of CoFeAl-LDH is affected in the electrolyte containing brine.

Response:

Thank you for pointing out this issue. As shown in Appendix Figure 2 and Supplementary Table 5, the composition of seawater is highly complex. During electrolysis, not only Cl⁻ and Br⁻ can poison the electrodes, but other species present may also affect the performance of the electrodes. In order to verify whether the OER performance of CoFeAl-LDH is affected in the electrolyte containing brine, the OER performance of CoFeAl-LDH was evaluated in 20 wt.% NaOH containing saturated NaCl and 6-fold concentrated seawater, respectively (Supplementary Figure 34). The results indicate that there is no significant change in the onset overpotential of CoFeAl-LDH when using the electrolyte containing 6-fold concentrated seawater. The difference in performance is mainly reflected in the current growth rate, which may be attributed to the enhanced adsorption of Cl⁻/Br⁻ ions at higher potentials.

Appendix Figure 2. The concentration of ions in real seawater and 6-fold concentrated seawater.

Revisions made in the Manuscript:

“The components of the brine were analyzed using ion chromatography and ICP-OES (Figure 2d and Supplementary Table 5).”

“And the OER performance of CoFeAl-LDH in this electrolyte was first confirmed to be unaffected significantly (Supplementary Figure 34).”

Supplementary Table 5. The concentration of ions in 6-fold concentrated seawater and real seawater.

Composition	6x concentrated seawater	Real seawater	Impact
Cl ⁻	119859.36 ppm	19365.75 ppm	Negative

Na ⁺	61302.81 ppm	10765.89 ppm	No impact
Br ⁻	1045.815 ppm	171.254 ppm	Negative
SO ₄ ²⁻	9384.135 ppm	2712.55 ppm	Positive
K ⁺	2006.51 ppm	433.32 ppm	No impact
Ca ²⁺	822.765 ppm	418.97 ppm	Negative
Mg ²⁺	7019.895 ppm	1299.97 ppm	Negative

Supplementary Figure 34. The polarization curves of CoFeAl-LDH in 20wt.% NaOH containing saturated NaCl and 6-fold concentrated seawater, respectively.

3. *For the first time, the authors have proposed hydrogen bonding network to repelled Cl⁻, please give more discussion on this mechanism and analyze whether there are other factors that may influence the corrosion resistance ability.*

Response:

Thank you very much. In this work, CoFeAl-LDH is etched by OH⁻ in the alkaline electrolyte and adsorbed on the electrode surface, forming a distal coordination structure. These distal Al³⁺ ions accumulate OH⁻ and coordinate with them to form Al(OH)_n⁻. Since both Al(OH)_n⁻ and the accumulated OH⁻ have negative charge, they repel the negatively charged Cl⁻ through electrostatic forces. Moreover, OH⁻ is considered one of the contributors of the hydrogen bond network, which facilitates the Grotthuss migration of other OH⁻ in the electrolyte toward the electrode surface. This process enhances the coverage of OH⁻ at the electrode surface, consequently mitigating Cl⁻ corrosion.

Regarding “the other factors” you mentioned, previous reports have mainly concentrated on investigating electrostatic repulsion. Several studies have indicated that the intensity of electrostatic repulsion is correlated with the extent of charge localization of ions (*J. Energy Chem* **2022**, *72*, 361–369), meaning that ions with smaller radii and higher charge possess the ability to exert more robust repulsive forces on Cl^- . Our previous research indicates that anions with larger radii can impede the movement of Cl^- towards the electrode surface via steric repulsion (*Angew. Chem.* **2023**, *135*, e202309882). Additionally, some studies have shown that the addition of anions in the electrolyte can also have a repulsive effect on Cl^- (*Angew. Chem.* **2021**, *133*, 22922–22926.; *Proc. Natl. Acad. Sci. U.S.A.* **2019**, *116*, 6624–6629). Unlike the ions generated from the electrode reconstruction, anions in the electrolyte need to undergo electro migration to reach the electrical double layer of the anode. Hence, the migration rate of ions may also have a certain influence.

Reply to Reviewer 2

Compared to the maturity of desalination technology, direct seawater electrolysis is often criticized for its high cost and continuous operation challenges. However, this work not only demonstrates the feasibility of low-cost, high-stability direct seawater electrolysis, but also proposes a possible solution to the saline water concentration problem caused by seawater desalination. In addition to promoting academic research on seawater electrolysis, this work has the potential to significantly propel industrialization in this field. To further improve this manuscript, I have the following suggestions.

Response:

We are grateful for the time and effort the reviewer had spent on our manuscript, and deeply appreciate the valuable review and positive comments of our manuscript, which have helped improve the manuscript quality.

- 1. It is noteworthy that the brine employed by the authors is 6X concentrated seawater. Please supply the composition analysis of the seawater before concentration and explain the possible impact of various substances present on seawater electrolysis.*

Response:

Thank you for pointing this issue. As shown in Appendix Figure 2. and Supplementary Table 5, the composition of seawater is exceedingly complex. Notably, the presence of Ca^{2+} and Mg^{2+} ions pose a challenge due to their propensity to precipitate upon the introduction of NaOH , necessitating their prior removal. Furthermore, during the process of electrolysis, Cl^- and Br^- ions act synergistically to induce severe corrosion on the anode. This corrosive effect intensifies with the concentration of seawater, leading to more severe corrosion. In addition, seawater also contains small amounts of ions such as SO_4^{2-} and CO_3^{2-} . When these ions adsorb onto the anode, they can enhance its stability.

Appendix Figure 2. The concentration of ions in real seawater and 6-fold concentrated seawater.

Revisions made in the Manuscript:

“The components of the brine were analyzed using ion chromatography and ICP-OES (Figure 2d and Supplementary Figure 5).”

Supplementary Table 5. The concentration of ions in 6-fold concentrated seawater and real seawater.

Composition	6x concentrated seawater	Real seawater	Impact
Cl ⁻	119859.36 ppm	19365.75 ppm	Negative
Na ⁺	61302.81 ppm	10765.89 ppm	No impact
Br ⁻	1045.815 ppm	171.254 ppm	Negative
SO ₄ ²⁻	9384.135 ppm	2712.55 ppm	Positive
K ⁺	2006.51 ppm	433.32 ppm	No impact
Ca ²⁺	822.765 ppm	418.97 ppm	Negative
Mg ²⁺	7019.895 ppm	1299.97 ppm	Negative

2. *In industry, alkaline water electrolyzers typically operate at current densities below 1 A cm⁻². And the MEA used by the authors can stably operate at 1 A cm⁻² even when using 6X concentrated seawater as the electrolyte, which is encouraging. I think it would be valuable if the authors could do a costing account based on this electrolyzer.*

Response:

Thank you very much for your valuable comment. Alkaline water electrolyzers have undergone decades of development, resulting in a highly mature technology. Appendix Table 1 presents the specifications for both alkaline water electrolyzer (AWE) and seawater electrolyzer (SWE), and Appendix Table 2 outlines the materials employed. The data for alkaline water electrolyzer is sourced from the literature "*International Journal of Hydrogen Energy* **2023**, 48, 32313–32330", while the data for seawater electrolyzer originates from our small-scale device (Supplementary Figure 25).

Based on the specifications (Appendix Table 2), it is evident that both types of electrolyzers operate at the same temperature and pressure. However, at similar voltages, the current density achieved by SWE is more than four times that of AWE. Compared to AWE's power density of only 0.45 W/cm², SWE boasts a significantly higher power density of 2.06 W/cm², indicating its superior performance. Additionally, due to the higher current density, SWE requires fewer cells when producing the same amount of hydrogen with electrodes of the same size. This will result in a smaller footprint for the constructed electrolysis equipment.

According to the materials employed (Appendix Table 2), AWE and SWE utilize the same cathode and the same diaphragm. Compared to AWE, the electrolyte used in SWE has a lower cost. This is because SWE only requires simple removal of Ca²⁺ and Mg²⁺ ions from seawater, whereas AWE necessitates at least a reverse osmosis process to produce ultrapure water.

However, the use of a CoFeAl-LDH anode, titanium alloy bipolar plates, and titanium alloy end plates in SWE increases the overall cost of the electrolyzer. It should be noted that recent advancements in corrosion prevention technology are expected to lead to the development of new, inexpensive materials for bipolar plates and end plates, significantly reducing the material cost of SWE.

Appendix Table 1. Electrolyzer specifications for Alkaline water electrolyzer and Seawater electrolyzer

	Alkaline water electrolyzer ^[1]	Seawater electrolyzer
Current density	0.245 A/cm ²	1.0 A/cm ²
Voltage	1.85 V	2.06 V
Power density	0.45 W/cm ²	2.06 W/cm ²
Temperature	80 °C	80 °C

Pressure

Ambient

Ambient

Appendix Table 2. Electrolyzer specifications for Alkaline water electrolyzer and Seawater electrolyzer

	Alkaline water electrolyzer [1]	Seawater electrolyzer
Anode	Ni foam	CoFeAl-LDH
Cathode	Ni foam	Ni foam
Separator	Zirfon UTP 500	Zirfon UTP 500
Electrolyte	Ultrapure water containing 20wt.% NaOH	Ca, Mg-free seawater containing 20wt.% NaOH
Bipolar plate	Ni plated Carbon Steel	Titanium alloy
End plate	Ni plated Carbon Steel	Titanium alloy

[1] Krishnan, S. *et al.* Present and future cost of alkaline and PEM electrolyser stacks. *International Journal of Hydrogen Energy* **48**, 32313–32330 (2023).

3. *Chlor-alkali electrolyzers are also widely used for brine utilization, enabling simultaneous production of H₂, Cl₂, and NaOH. A comparative analysis needs to be conducted to assess their respective advantages and disadvantages.*

Response:

Thank you very much. The usage scenarios of the two types of electrolyzers represent one of the most crucial limiting conditions for determining their respective advantages and disadvantages. From the perspective of revenue generation. It cannot be denied that in addition to producing hydrogen, the chlor-alkali industry also generates high-value Cl₂ (or NaClO) and NaOH, resulting in significantly higher economic benefits compared to the seawater electrolysis route for hydrogen production.

However, from the perspective of producing clean energy. It is crucial to consider the high toxicity and corrosiveness of the products generated by chlor-alkali industry, which necessitates their local consumption. Any surplus production can easily cause environmental pollution. Conversely,

Concentrated brine electrolysis for hydrogen production by adjusting the pH to alkaline is an environmentally friendly route.

Moreover, membrane-based desalination technologies can lead to the concentration of brine (Appendix Figure 3), which poses a significant threat to the marine ecological environment when discharged into the water body. Taking the common reverse osmosis combined with electro dialysis as an example (RO+ED), the produced concentrated brine contains 4–5 M NaCl (Appendix Figure 4). If all these concentrated brines are used for chlor-alkali industry to produce Cl₂ (or NaClO) and NaOH, it will inevitably lead to excess production, which can cause environmental pollution. Therefore, from the perspective of concentrated brine disposal, seawater electrolyzers also have significant advantages.

Appendix Figure 3. Volume of brine produced per country.

Appendix Figure 4. Concentration process of brine in RO+ED desalination route.

- In summary, the progress achieved in this work is important in the field of seawater electrolysis, and I highly recommend its publication in Nature Communications.*

Response:

We express our gratitude for the reviewer's dedicated time and effort in providing feedback on our manuscript.

Reply to Reviewer 3

This study developed an CoFeAl layered double hydroxide (CoFeAl-LDH) anodes, which shows excellent oxygen evolution reaction (OER) activity and stability. The Al^{3+} of CoFeAl-LDH in electrode were etched off by OH^- during OER process, resulting in M^{3+} vacancies that boost OER activity. Additionally, the self-originated $Al(OH)_n$ was found adsorb on the anode surface to improve stability. However, before this paper could be considered for publication, some main issues should be well addressed.

Response:

Thank you very much for your unreserved effort in reviewing our manuscript. Your comments are highly insightful and enable us to greatly improve the manuscript. We have revised the manuscript carefully according to your suggestions. Details of the corresponding revisions are described below point by point.

- 1. The authors should describe the peaks that appear in the LSV curves of Figures 2a and 2b and explain why the peaks in Figures 2a and 2b are not in the same position.*

Response:

Thank you for pointing this issue. The peaks observed at 1.3 V vs RHE in Figure 2a and 1.22 V vs RHE in Figure 2b are attributed to the reduction peaks generated by the redox-active Ni (in the substrates) and Co (in the nanoarrays) elements. Taking $CoOOH/Co(OH)_2$ as an example, the reaction equation is as follows: $CoOOH + H_2O + e^- \rightleftharpoons Co(OH)_2 + OH^-$.

The position of the redox peaks is correlated with the concentration of OH^- in the electrolyte, as demonstrated in Supplementary Figure 36, which shows the relationship between these two factors. It can be observed that with the gradual increase in NaOH concentration in the electrolyte from 1 M to 6 M (approximately 20 wt.%), the redox peak positions shift towards lower potentials. This phenomenon indicates that the reaction $CoOOH + H_2O + e^- \rightleftharpoons Co(OH)_2 + OH^-$ becomes more difficult to occur due to the influence of OH^- concentration on the reaction equilibrium.

As the concentration of OH^- increases, the reaction equilibrium shifts in a direction that favors the formation of $CoOOH$, making it more difficult for the reverse reaction to occur. This is consistent with the observed shift of the redox peaks to lower potentials as the OH^- concentration increases. It is important to note that the potentials at NaOH concentrations of 1 M and 6 M are consistent with those presented in Figure 2a and Figure 2b, respectively.

In addition, we also analyzed the onset potential of OER. It is evident that the onset potential decreases with the NaOH concentration increase. This observation indicates that variations in OH⁻ concentration impact the reaction rate and charge transfer processes at the electrode surface.

The reduction peaks in Figure 2b were also analyzed, showing that CoFeAl-LDH demonstrates a slightly lower potential compared to the CoFe-LDH. This phenomenon suggests that the formation of high-valent sites is more facile in CoFeAl-LDH. Moreover, the increased peak area of CoFeAl-LDH indicates a greater amount of such sites, which is related to the dissolution of Al³⁺.

Revisions made in the Manuscript:

“Note that the peaks observed at 1.3 V vs RHE in Figure 2a and 1.22 V vs RHE in Figure 2b are attributed to the reduction peaks generated by the redox-active Ni and Co elements in the material. And the position of the redox peaks is correlated with the concentration of OH⁻ in the electrolyte (Supplementary Figure 36).”

Supplementary Figure 36. Cyclic voltammetry curves of CoFeAl-LDH in the electrolytes containing different NaOH concentration.

- In Figure 3g, the after-activation CoFeAl-LDH still maintains a sheet-like morphology of LDH, but the XRD of the after-activation CoFeAl-LDH in Figure 3a no longer shows peaks corresponding to LDH sheets. The author should provide an explanation for this discrepancy.*

Response: Thank you very much for your valuable comment. During the OER process, CoFeAl-LDH undergoes reconstruction, resulting in the formation of substances such as Co(Fe,Al)OOH with low crystallinity. However, when the voltage is no longer applied, Co(Fe,Al)OOH undergo another reconstruction process back into CoFeAl-LDH. This process leads to a decrease in the

crystallinity of the material, which is the main reason for the disappearance of peaks attributed to LDH in XRD pattern. And the phenomenon of minimal morphological changes alongside changes in the crystal structure of materials is termed as “topological transformation”. This strategy is often employed in the fabrication of nanomaterials like MP_x , MS_x , MSe_x , MO_x . (*Nat. Commun.* **2018**, *9*, 924; *Research* **2020**, *2020*, 1–9; *J. Mater. Chem. A* **2016**, *4*, 13731–13735.)

At first, the reconstruction process was monitored using Raman spectroscopy. It is important to note that the dissolved $Al(OH)_n^-$ during the reconstruction of CoFeAl-LDH may affect the signal of Co(Fe)OOH. Therefore, CoFe-LDH was used during this step. As shown in Figure Appendix Figure 5, once the applied voltage reached above 1.3 V vs RHE, the sample exhibited M-O vibration signals attributed to Co(Fe)OOH (*Energy Environ. Sci.* **2022**, *15*, 727–739). This phenomenon can also be observed in Appendix Figure 6. Moreover, when the applied voltage is removed, the Raman spectrum once again exhibits features characteristic of CoFe-LDH (Appendix Figure 6).

Appendix Figure 5. Operando Raman spectrum of CoFe-LDH, potential range: 1.2-1.5 V vs RHE.

Appendix Figure 6. Raman spectrum of CoFe-LDH (before activation), Co(Fe)OOH (under 1.5 V vs RHE), and CoFe-LDH (after activation).

After demonstrating the reconstruction process of CoFeAl-LDH (before activation) \square Co(Fe,Al)OOH (under applied potential) \square CoFeAl-LDH (after activation), comparative analyses of the LDHs before and after the reaction were conducted using HAADF-STEM. As shown in Supplementary Figure 35, in contrast to the well-ordered lattice fringes observed in CoFeAl-LDH (before activation), the lattice fringes of CoFeAl-LDH (after activation) were found to be more disordered, with notable presence of amorphous regions. This phenomenon suggests a reduced crystallinity in CoFeAl-LDH (after activation).

Supplementary Figure 35. HAADF-STEM images of (a) CoFeAl-LDH before activation, and (b) CoFeAl-LDH after activation.

Regarding your mention of "the after-activation CoFeAl-LDH still maintains a sheet-like morphology of LDH," we have provided two additional SEM images at higher magnification in Appendix Figure 7. It can be observed that the nanosheets of CoFeAl-LDH (after activation) exhibit a wrinkled appearance with a rougher surface, which is also related to the electrode reconstruction.

Appendix Figure 7. Raman spectrum of (a) CoFe-LDH (before activation), (b) Co(Fe)OOH (under 1.5 V vs RHE), and CoFe-LDH (after activation).

Furthermore, the duration of activation is also related to the crystallinity of CoFeAl-LDH (after activation). As shown in Appendix Figure 8, an extended activation period leads to a gradual reduction in the intensity of the XRD diffraction peaks associated with LDH.

Appendix Figure 8. XRD of CoFeAl-LDH (before activation), CoFeAl-LDH (after 5 h activation), and CoFeAl-LDH (after 10 h activation).

Revisions made in the Manuscript:

“Additionally, as shown in the high-angle annular dark field scanning transmission electron microscope (HAADF-STEM) images (Supplementary Figure 35), the activated CoFeAl-LDH exhibits a reduced crystallinity, further corroborating the occurrence of reconstruction.”

Supplementary Figure 35. HAADF-STEM images of (a) CoFeAl-LDH before activation, and (b) CoFeAl-LDH after activation.

- In Figure 3d, regarding the XPS analysis of Co 2p, the peak area ratio of Co 2p_{3/2} to Co 2p_{1/2} for the same oxidation state should be 2:1, and the full width at half maximum (FWHM) should be consistent. However, it does not meet these requirements. The author should review the XPS peak decomposition rules for this element and make the necessary adjustments.*

Response:

Thank you for pointing out this issue. We have carefully studied the XPS peak decomposition rules from the reference "*J. Vac. Sci. Technol. A*, **2022**, *40*, 063201". Following the method you mentioned, we re-analyzed the Co 2p XPS data referencing the articles "*Appl. Surf. Sci.*, **2011**, *257*, 2717–2730; *Appl. Surf. Sci.*, **2015**, *351*, 1016–1024". These analyses were performed using the Aventure software (Thermo Fisher Scientific). The results are updated in Figure 3d and Supplementary Figure 10. Key parameters used for peak decomposition, including peak areas and FWHM, are summarized in Appendix Table 3.

Appendix Table 3. Peak area and FWHM for Co 2p XPS decomposition.

Sample	Peak	Area (CPS.eV)	FWHM (eV)
CoFeAl-LDH before activation	Co 2p _{3/2}	24383.14	3.50
	Co 2p _{1/2}	12635.95	3.50
	Co 2p _{3/2} sat.	18426.30	4.59
	Co 2p _{1/2} sat.	9466.30	4.59
CoFeAl-LDH after activation	Co 2p _{3/2}	8286.08	3.43
	Co 2p _{1/2}	4296.89	3.43
	Co 2p _{3/2} sat.	2895.14	3.46
	Co 2p _{1/2} sat.	1497.44	3.46
CoFe-LDH before activation	Co 2p _{3/2}	15750.69	3.50
	Co 2p _{1/2}	8162.40	3.50
	Co 2p _{3/2} sat.	10853.99	3.46
	Co 2p _{1/2} sat.	5612.45	3.46

Revisions made in the Manuscript:

35. Major, G. H. et al. Guide to XPS data analysis: Applying appropriate constraints to synthetic peaks in XPS peak fitting. *J. Vac. Sci. Technol. A* **40**, 063201 (2022).
36. Tholkappiyan, R. & Vishista, K. Tuning the composition and magnetostucture of dysprosium iron garnets by Co-substitution: An XRD, FT-IR, XPS and VSM study. *Appl. Surf. Sci.* **351**, 1016–1024 (2015).
37. Biesinger, M. C. *et al.* Resolving surface chemical states in XPS analysis of first row transition metals, oxides and hydroxides: Cr, Mn, Fe, Co and Ni. *Appl. Surf. Sci.* **257**, 2717–2730 (2011).

Figure 3d. Co 2p High-resolution XPS spectrum of CoFeAl-LDH before and after activation.

Supplementary Figure 10. Co 2p High-resolution XPS of CoFe-LDH and CoFeAl-LDH.

4. *In line 51, there is a numerical omission in "evaluation at A cm⁻²". The author should make the correction. Similar language errors can be observed elsewhere, so the author should pay attention to writing standards.*

Response:

Thank you very much for your valuable comment. We originally intended to express that "the currently reported anodes still lack stability **evaluation at ampere current densities**". However, the wording we used was not accurate. The corresponding sentence has now been revised. Additionally, we have reviewed the entire manuscript to prevent this issue from recurring. Thank you again for pointing this out, and we apologize for any confusion it may have caused. **Revisions**

made in the Manuscript:

"Furthermore, the currently reported anodes still lack of stability evaluation at ampere level current densities and saturated salinity levels."

"signifying a more pronounced corrosive effect of the brine-containing electrolyte on the electrode (the corrosion mechanism can be seen in Figure 1a and Supplementary Figure 2)."

“The analysis revealed that, in addition to containing ultra-high concentrations of Cl⁻ (119.86 g/L, equal to 1.20×10⁵ ppm),”

5. *Between lines 73-74, the correct conversion for 1 A cm⁻² should be 10000 A m⁻², while the text states 1000 A m⁻². The author should make the necessary correction and ensure thorough attention to detail in the writing to avoid such errors.*

Response:

Thank you for bringing this matter to our attention, and we apologize for any similar issues in our manuscript. The unit has been revised to "10 kA m⁻²" as per your suggestion. Furthermore, we have thoroughly reviewed the entire manuscript for unit conversions to uphold its precision and correctness. We appreciate your patience and valuable assistance.

Revisions made in the Manuscript:

“It is estimated that a seawater electrolysis system with a hydrogen production capacity of 500 Nm³ h⁻¹, operating at 10 kA m⁻² (equal to 1 A cm⁻²), will just take less than 3 days to reach electrolyte saturation with NaCl (Supplementary Figure 1).”

6. *The value of the overpotential in line 123 of the main text is incorrectly written.*

Response:

Thank you for your feedback. We appreciate your suggestion. The meaning we aimed to convey was that “CoFeAl-LDH demonstrates better OER performance compared to CoFe-LDH in an electrolyte with a high-concentration NaOH”. The “ $\Delta\eta_{10}$ ” in the manuscript represented “ $\eta_{10, \text{CoFe-LDH}} - \eta_{10, \text{CoFeAl-LDH}}$ ”. However, following your guidance, we acknowledge that this notation was not sufficiently clear. As a result, we have revised the sentence to eliminate any ambiguity.

Revisions made in the Manuscript:

“The OER performance and corresponding Tafel slope (Supplementary Figure 17) of CoFeAl-LDH and CoFe-LDH were recorded in Supplementary Figure 18, note that these two electrodes exhibited similar activity in low-concentration NaOH, but CoFeAl-LDH exhibited significantly better performance ($\Delta\eta_{10} = \eta_{10, \text{CoFe-LDH}} - \eta_{10, \text{CoFeAl-LDH}} = 23 \text{ mV}$) in high-concentration NaOH.”

7. *In Supplementary Figure 16, why are the current densities corresponding to each sample at a potential of 1.0 V not concentrated at the same point? The authors are requested to explain.*

Response:

Thank you very much. The presence of redox-active elements such as Ni (in the substrates) and Co (in the nanoarrays) leads to distinct reduction peaks at a voltage of 1.0 V, resulting in the current densities failed to concentrate at a single point. As shown in Appendix Figure 9, varying intensities of redox peaks are evident in the cyclic voltametric (CV) curves. To avoid interference from oxidation peaks in overpotential calculations, we have consistently employed negative scans throughout this manuscript.

Appendix Figure 9. (a) CV curves and (b) redox peaks of CoFeAl-LDH, CoFe-LDH, Commercial IrO₂ and Commercial Ni foam.

8. *The OH in line 294 of the text lacks the superscript "-".*

Response:

We apologize for any typos or errors in the manuscript, and we have carefully reviewed the entire document to rectify any similar issues. Thank you for your understanding and patience.

Revisions made in the Manuscript:

“Figure 5d presents a schematic diagram of the CoFeAl-LDH electrode how to repel Cl⁻ but admit OH⁻ simultaneously.”

9. *The subscript "n" is missing from Al(OH)⁻ in line 300 of the text.*

Response:

Thank you once again for your patience. We deeply apologize for these errors and have conducted a thorough word-by-word review of the entire manuscript to prevent similar mistakes from happening in the future. Your attention to detail is much appreciated.

Revisions made in the Manuscript:

“Thus, OH adsorption are selectively enhanced near to the electrode surface with Al(OH)_n coverage.”

Finally, we would like to express our sincere gratitude to the Editor and Reviewers again for the valuable comments and constructive suggestions to improve the quality of our manuscript. Thank you very much!

REVIEWERS' COMMENTS

Reviewer #1 (Remarks to the Author):

After carefully reading the revised version, I think now this manuscript is good enough to be accepted without further revision.

Reviewer #2 (Remarks to the Author):

I appreciate the authors' efforts devoted in revising the manuscript and all of my comments have been well addressed. I suggest the current form can be accepted for publication.

Reviewer #3 (Remarks to the Author):

The authors have addressed all the issues. It can be considered as a publication in the present form.